# FedBiF: Communication-Efficient Federated Learning via Bits Freezing

## Abstract

Federated learning (FL) is a promising privacy-preserving distributed machine learning paradigm, however, involves significant communication overhead. The communication latency of iterative model transmissions between the central server and the clients seriously affects the training efficiency. Recently proposed algorithms quantize the model updates to reduce the FL communication costs. Yet, existing quantization methods only compress the model updates after local training, which introduces quantization errors to the model parameters and inevitably leads to a decrease in model accuracy. Therefore, we suggest restricting the model updates within lower quantization bitwidth during local training. To this end, we propose **Federated Bits Freezing** (FedBiF), a novel FL framework that enables clients to train only partial individual bits inside each parameter, termed activated bits, while freezing the others. In this way, the model updates are restricted to the representation of activated bits during local training. By alternately activating each bit in different FL rounds, FedBiF achieves extremely efficient communication, as only one activated bit is trained for each parameter and subsequently transmitted. Extensive experiments are conducted on three popular datasets with both IID and Non-IID settings. The experimental results not only validate the superiority of FedBiF in communication compression but also reveal some beneficial properties of FedBiF, including model sparsity and better generalization. In particular, FedBiF outperforms all the baseline methods, including FedAvg, by a large margin even with 1 bit per parameter (bpp) uplink and 4 bpp downlink communication.

## 1 Introduction

Driven by the increasing data privacy concerns, federated learning (FL) McMahan et al. (2017); Mills et al. (2020) has emerged as a promising distributed machine learning paradigm, which allows distributed clients to collaboratively train a global model without accumulating their raw data to a centralized repository. The FL process is composed of a series of local training, parameters uploading, centralized aggregation, and parameters downloading phases, which will be repeated for several rounds until the model converges.

However, the model transmissions between the server and clients can significantly affect the training efficiency, especially for edge clients that are often of limited uplink and downlink bandwidth. As a result, communication-efficient federated learning has gained great research attention recently and numerous techniques are adopted to reduce the communication overhead Caldas et al. (2018); Jiang et al. (2019); Hyeon-Woo et al. (2022); Reisizadeh et al. (2020). In this paper, we investigate the application of quantization techniques in communication compression for FL.

Quantization is to represent the model parameters with fewer bits, thereby reducing the data volume for FL model synchronization. FedPAQ Reisizadeh et al. (2020) is the pioneer in utilizing quantization on model updates to improve communication efficiency for FL. Subsequent work Jhunjhunwala et al. (2021); Qu et al. (2022); Hönig et al. (2022) further improves the compression efficiency by adaptively adjusting the quantization bitwidth used in different rounds. AdaQuantFL Jhunjhunwala et al. (2021), for instance, employs time-adaptive quantization, using a lower bitwidth in the initial rounds and gradually increasing the bitwidth as training progresses. The authors of AdaQuantFL believe that early model updates do not need to be particularly refined. DAdaQuantHönig et al. (2022) uses similar time-adaptive quantization and proposes client-adaptive quantization, employing higher

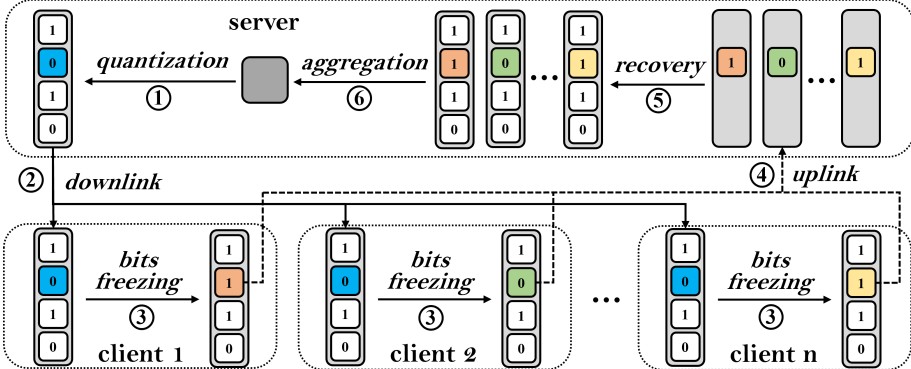

Figure 1: The pipeline of FedBiF. ①: server quantizes the model parameters; ②: server sends the 4-bit quantized parameters to clients; ③: clients train on the activated bit (the blue one) by bits freezing; ④: clients upload the updated activated bit (1 bit) to server. ⑤: server recovers model parameters with activated bits and frozen bits for each client. ⑥: server aggregates the recovered parameters.

bitwidth for clients with more data. The authors of DAdaQuant suggest that the clients with larger datasets should provide more precise model updates, as their updates make greater contributions to the global model during server aggregation. In contrast, FedDQ Qu et al. (2022) adopts a descending quantization strategy, gradually decreasing the bitwidth throughout the training process. The rationale behind FedDQ is that as the model progressively converges over training, the range of model updates will be smaller, enabling the utilization of lower bitwidth to effectively represent the local updates.

We argue that aforementioned bitwidth-adaptive quantization methods suffer from two major drawbacks, despite their achievements in reducing communication overhead for FL. Firstly, they just compress the model updates from clients (i.e., uplink) while leaving the global updates from the server to clients (i.e., downlink) uncompressed. In this paper, we seek to compress both uplink and downlink communication to further improve the communication efficiency. Secondly and most importantly, existing methods quantize the model updates in a post-training manner. Specifically, quantization occurs after local training and can not be aware by the training process. The post-training quantization will bring quantization errors to the parameters, resulting in loss of information for local model updates. In cases where the quantization bitwidth is extremely small (e.g., 1 bit), the quantization errors can be significant and will lead to a noticeable decrease in model accuracy. It is impossible to completely eliminate the quantization errors in model parameters, regardless of how we adjust the bitwidth. Therefore, it is more worthwhile to investigate how to get rid of the need for post-training quantization in FL. In this paper, we suggest restricting the model updates within lower bitwidth during local training, thus requiring no post-training quantization.

According to the above analysis, we propose **Federated Bits Freezing** (FedBiF), a novel framework that achieves bidirectional communication compression and enables clients to restrict model updates into lower bitwidth during local training. The pipeline of FedBiF is shown in Figure 1. In each round of FedBiF, the server first quantizes the global model and sends the quantized parameters to clients for efficient downlink communication. To avoid post-training quantization, we seek to keep the quantized parameters within the representation range of their original bitwidth during training. To this end, we propose to update the individual bits inside each quantized parameter, rather than the parameter itself. The key idea of FedBiF is to train and communicate only partial bits inside the quantized parameters, termed activated bits, while freezing the others. In this way, the model updates can be represented by the activated bits which are learned during local training. By alternately activating different bits in different FL rounds, FedBiF achieves effective training on parameters and efficient communication simultaneously. After local training, without quantization, clients just need to upload the activated bits to the server where the local models will be recovered and aggregated afterward.

Experimentally, we find that FedBiF not only is quite effective in communication compression, but also is capable of some beneficial model properties, including model sparsity and better generalization. FedBiF can automatically prune a parameter when the bits inside the parameter are all optimized into zero. In our experiments, FedBiF achieves an average sparsity of about 30% on different models.

Also, we analyze that FedBiF achieves better generalization by limiting the growth of the model's Lipschitz constantWeng et al. (2018). Specifically, FedBiF demonstrates a 2∼9% improvement in test accuracy when compared to FedAvg. We will discuss these properties in detail in Section 4.4.

The contributions are summarized as follows:

- We dispute the post-training manner of existing federated quantization methods, which causes accuracy loss when applying lower quantization bitwidth. We propose to restrict the model updates into fewer bits during local training, which opens new avenues for communication-efficient FL.
- We propose Federated Bits Freezing (FedBiF), a novel FL framework that enables clients to train and communicate only partial activated bits inside a parameter. FedBiF restricts the model updates into the representation of activated bits during local training, requiring no post-training quantization.
- Extensive experiments are conducted on three public datasets. The experimental results reveal not only the superiority of FedBiF in bidirectional communication compression but also its properties of model sparsity ad better generalization. FedBiF can outperform all baselines, including FedAvg, by a large margin even with 1 bit per parameter (bpp) uplink and 4 bpp downlink communication.

## 2 PRELIMINARIES

### 2.1 FEDERATED LEARNING

Federated learning is built on a client-server topology in which $N$ clients are connected to a central server. Each client $C_k$ owns a local datasets $D_k$ and we denote $p_k = |D_k|/\sum_j |D_j|$ as the proportion of its data quantity to the total data quantity. The general goal of federated learning is to train a global model by multiple rounds of local training on each client's local datasets. Denoting the model parameters as $\mathbf{w} \in \mathbb{R}^d$, where $d$ is the dimension of the model parameters, the objective function of client $C_k$ as $f_k$, federated learning can be formulated as $\min_{\mathbf{w} \in \mathbb{R}^d} F(\mathbf{w}) := \sum_{k=1}^{N} p_k f_k(\mathbf{w})$.

FedAvg McMahan et al. (2017) is a widely used FL algorithm. In the $t_{th}$ round, the server sends the model parameters $\mathbf{w}^t$ to several randomly selected clients, denoted as $\mathbb{C}^t$. Each selected client $C_k$ performs certain epochs of local training and sends the updated model parameters $\mathbf{w}_k^{t+1}$ back to server. The server aggregates these updated model to generate a new global model as $\mathbf{w}^{t+1} = \sum_{\mathbb{C}^t} p_k \mathbf{w}_k^{t+1}$.

### 2.2 STOCHASTIC QUANTIZATION

Quantization is to represent full-precision parameters with fewer bits. In this paper, we adopt stochastic quantization Krishnamoorthi (2018) on model parameters and model updates following existing worksReisizadeh et al. (2020); Qu et al. (2022); Li et al. (2023). Specifically, given the quantization step size $\alpha$ and the quantization bitwidth $m$, a full-precision tensor $\mathbf{x}$ will be quantized element-wise into a $m$-bit integer tensor $\bar{\mathbf{x}}$ for communication compression:

$$\bar{\mathbf{x}} = Q(\mathbf{x}, \alpha, m) = round(clamp(\mathbf{x}/\alpha, -2^{m-1}, 2^{m-1} - 1)), \tag{1}$$

where $clamp(x, a, b)$ limits $x$ into the range of $[a, b]$ and $round(x)$ rounds $x$ to its two adjacent integers with a probability distribution:

$$round(x) = \begin{cases} \lfloor x \rfloor & \text{w.p.} & \lfloor x \rfloor + 1 - x, \\ \lfloor x \rfloor + 1 & \text{w.p.} & x - \lfloor x \rfloor. \end{cases} \tag{2}$$

After communication, the integer tensor $\bar{\mathbf{x}}$ will be de-quantized into floating-point values $\hat{\mathbf{x}}$ as:

$$\hat{\mathbf{x}} = DeQ(\bar{\mathbf{x}}, \alpha) = \alpha \cdot \bar{\mathbf{x}}. \tag{3}$$

## 3 METHODOLOGY

As described in Figure 1 about FedBiF, the server sends the quantized global model to clients for efficient downlink communication. The clients perform bits freezing, training only several activated bits and freezing the others. In this section, we first introduce how clients perform bits freezing in Section 3.1 and then give a detailed description of the proposed framework FedBiF in Sectoion 3.2.

## 3.1 BITS FREEZING

In the $t_{th}$ round, clients receive $m$-bit quantized model parameters $\bar{\mathbf{w}}^t = \{(\bar{\theta}_l^t, \alpha_l^t)|l = 1..L\}$, where $\bar{\theta}_l^t$ and $\alpha_l^t$ are the integer parameters and the step size of the $l_{th}$ layer. For each layer, we decompose its integer parameters $\bar{\theta}_l^t$ into a weighted sum of $m$ bits $\{b_{m-1,l}^t..b_{1,l}^t, b_{0,l}^t\}$:

$$\bar{\theta}_l^t = \sum_{i=0}^{m-1} 2^i \cdot b_{i,l}^t - 2^{m-1}. \tag{4}$$

All bits are utilized as magnitude bits. Instead of maintaining a sign bit, we obtain signed integers by subtracting $2^{m-1}$. The drawbacks of training a sign bit shall be discussed in Section 4.3.

The key of bits freezing is to train and update only several activated bits inside the model parameters. Given the binary values of $m$ bits within the model parameters, we first calculate their integer values by Eq. 4 and then get the floating-point parameters by Eq. 3. The floating-point parameters will be used to calculate model outputs and obtain gradients for optimizing the activated bits. However, the values of individual bits are discrete 0 or 1, which are infeasible to be optimized by gradient descent algorithms. Inspired by Binary Neural Networks Courbariaux et al. (2015), we maintain a real-valued virtual bit for each binary bit. In the forward pass, we perform a step function ($h(x) = 1$ if $x > 0$, otherwise 1) on the virtual bits to get the binary bit values. In the backward pass, the virtual bits will be optimized by straight-through estimator (STE) Hinton (2012), which considers the step function as an identity projection ($\partial h(x)/\partial x = 1$) when computing the gradients.

## 3.2 FEDERATED BITS FREEZING

---

**Algorithm 1: Federated Bits Freezing**

---

**Input:** $T$: communication round; $E$: local epoch; $N$: number of clients per round;
$\qquad$ $m$: quantization bitwidth; $\mathcal{S}$: bit selector.
Server initializes global model parameters $\mathbf{w}^0$.
Clients initialize virtual bits with Kaiming initialization.
**for** $t = 0$ *to* $T - 1$ **do**
$\quad$ Server randomly selects $N$ clients $\mathbb{C}^t = \{C_0, C_1, ..., C_{N-1}\}$.
$\quad$ Server quantizes global model parameters $\mathbf{w}^t$ to $m$ bits by Eq. 5.
$\quad$ Server broadcasts the quantized model to the clients in $\mathbb{C}^t$.
$\quad$ **for** $C_k \in \mathbb{C}^t$ **in parallel do**
$\quad\quad$ **for** $e = 0$ *to* $E - 1$ **do**
$\quad\quad\quad$ Client reinitializes local model by virtual bits inheritance.
$\quad\quad\quad$ Client trains the activated bits selected by $\mathcal{S}$.
$\quad\quad$ **end**
$\quad\quad$ Client sends the activated bits to the server.
$\quad$ **end**
$\quad$ Server performs aggregation to get $\mathbf{w}^{t+1}$ by Eq. 6.
**end**

---

In this subsection, we introduce the proposed bits freezing technique into federated learning settings to improve communication efficiency. The complete workflow of FedBiF is summarized in Algorithm 1.

In FedBiF, the model architecture on the server remains the same as usual, while the architecture of local models is designed for local training with bits freezing, where a weight is replaced by several virtual bits. Prior to training, the virtual bits of local models are initialized with Kaiming initialization He et al. (2015). In each round, the server first sends the quantized global model to clients. Given the bitwidth $m$, the $l_{th}$ layer of the global model $\theta_l$ will be quantized using Eq. 1 as:

$$\bar{\theta}_l = Q(\theta_l, \alpha_l, m), \quad \text{where } \alpha_l = ||\theta_l||_\infty/2^{m-1}. \tag{5}$$

Upon receiving the quantized model, clients can decompose the global integer parameters using Eq. 4 and easily get the binary bit values. Next, each client updates its local model by reinitializing the

virtual bits following a simple strategy, called virtual bits inheritance (VBI). Specifically, the value of each binary bit indicates only the sign of its corresponding virtual bit and not the magnitude. In order to preserve local model features, VBI reuses the magnitudes of the original virtual bits and sets their signs according to the binary bit values. The significance of VBI will be discussed in Section 4.3.

After reinitialization, each client trains on some activated bits chosen by a bit selector. In this paper, we adopt a simple yet effective selector that traverses all bits without overlap, activating $s$ bits at a time, where $s$ is the given number of activated bits. For example, if the bandwidth is 4 and 2 bits are activated in each round, FedBiF iterates over two bit patterns: '1100' and '0011', where '1' denotes the bit is activated. All clients share the same bit pattern to ensure similar magnitude of model updates. After local training, clients send their activated bits to the server where the local model can be recovered and then aggregated. In fact, the model recovery can be omitted as we can get aggregated model by adding the frozen bits to the aggregated results of the activated bits. For example, assuming the subset of activated bits is $\mathbb{S}^t$ and the subset of selected clients is $\mathbb{C}^t$, the aggregation of the $l_{th}$ layer can be formulated as:

$$\theta_l^{t+1} = \alpha_l^t \cdot \left( \frac{1}{|\mathbb{C}^t|} \sum_{k \in \mathbb{C}^t} \sum_{i \in \mathbb{S}^t} 2^i \cdot b_{i,l,k}^{t+1} + \sum_{j \notin \mathbb{S}^t} 2^j \cdot b_{j,l}^t - 2^{m-1} \right). \tag{6}$$

## 4 EXPERIMENTS

### 4.1 EXPERIMENTAL SETUP

**Datasets and Models.** In this paper, we evaluate FedBiF on three popular datasets in image classification tasks: SVHN Netzer et al. (2011), CIFAR-10 and CIFAR-100 Krizhevsky & Hinton (2009). However, it is worthy noting that FedBiF is generally applicable to other tasks, such as natural language processing. To demonstrate the versatility of FedBiF, we evaluate it across different model architectures. In particular, we adopt a lightweight version VGG11 Simonyan & Zisserman (2015) for SVHN, ResNet-18 He et al. (2016) for CIFAR-10 and ResNet-34 He et al. (2016) for CIFAR-100. Further information regarding the datasets and models can be found in Appendix A.1.

**Data Partition.** Here, we consider both cases of IID and Non-IID data partition to comprehensively showcase the performance. We partition data according to the Non-IID data partitioning benchmark Li et al. (2022). For IID partition, we randomly sample the same amount of data for each client. For the Non-IID case, we use label ratios generated by the Dirichlet distribution Yurochkin et al. (2019) and get varying degrees of Non-IID data by setting the Dirichlet parameter $\mu$ to 0.5 and 0.1, respectively.

**Baseline Methods.** FedAvg is adopted as the backbone algorithm in our experiments. We further compare FedBiF with several state-of-the-art quantization methods, including SignSGD Bernstein et al. (2019), FedPAQ Reisizadeh et al. (2020), FedDQ Qu et al. (2022) and DAdaQuant Hönig et al. (2022). SignSGD only transfers the sign of the model updates and aggregates the binary updates by majority vote. FedPAQ is to quantize the model updates by a given quantization bitwidth. FedDQ is to quantize the model updates with a specific step size, so that the quantization bitwidth gets smaller as the range of model updates decreases. DAdaQuant dynamically adjusts the bitwidth utilized by each client in each round by monitoring the local training loss and the local dataset size. More details about the baseline methods can be found in Appendix A.2.

**Hyperparameters.** In our experiments, the number of clients is set to 10. All clients participate in every round to eliminate the effect of randomness brought by party sampling McMahan et al. (2017). The batch size is set to 64 and the number of local epoch is set to 5. SGD Bottou (2010) is used as the local optimizer and the learning rate is tuned from (0.1, 0.01, 0.001). For SVHN, we set the learning rate to 0.01 and run 100 rounds. For CIFAR-10 and CIFAR-100, we set the learning rate to 0.1 and run 200 rounds. As for the baselines, each hyperparameter is carefully tuned. For FedPAQ and FedDQ, we tune the quantization bitwidth or the step size to achieve the highest compression without sacrificing accuracy. For SignSGD and DAdaQuant, we conduct experiments according to the settings in the original papers Bernstein et al. (2019); Hönig et al. (2022). For FedBiF, the quantization bitwidth and the number of activated bits are set to 4 and 1 by default. Detailed tuning process and results are shown in Appendix A.2. Each experiment is run five times on Nvidia 3090 GPUs with Intel Xeon E5-2673 CPUs. The average results and standard deviation are reported.

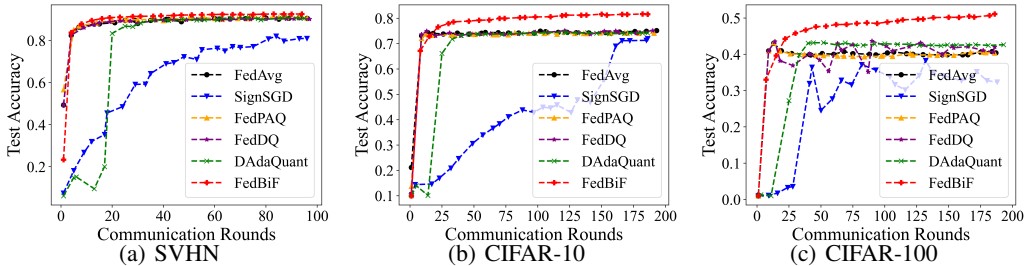

Figure 2: Convergence curves of baseline methods and FedBiF on IID datasets.

## 4.2 PERFORMANCE EVALUATION

In this subsection, we compare FedBiF with the baseline methods by three metrics including test accuracy, uplink and downlink communication costs. All the numerical results are reported in Table 1. The convergence curves of all methods on IID datasets are shown in Figure 2. The convergence curves on Non-IID datasets look similar and can be found in Appendix B.1.

Table 1: The test accuracy and communication costs of baselines and FedBiF. The total uncompressed uplink and downlink communication costs are reported in the shaded area. The average bit per parameter (bpp) for communication of different methods are shown in the corresponding columns.

| | IID | | Non-IID ($\mu = 0.5$) | | Non-IID ($\mu = 0.1$) | | |
|---|---|---|---|---|---|---|---|
| | Accuracy | Uplink | Accuracy | Uplink | Accuracy | Uplink | Downlink |
| SVHN with VGG11 (2.6 GB) | | | | | | | |
| FedAvg | 90.9 ($\pm$ 0.3) | 32 bpp | 86.3 ($\pm$ 0.3) | 32 bpp | 73.5 ($\pm$ 0.3) | 32 bpp | 32 bpp |
| SignSGD | 82.5 ($\pm$ 0.8) | 1 bpp | 80.8 ($\pm$ 0.6) | 1 bpp | 64.5 ($\pm$ 1.0) | 1 bpp | 1 bpp |
| FedPAQ | 91.0 ($\pm$ 0.2) | 4 bpp | 86.1 ($\pm$ 0.2) | 4 bpp | 73.9 ($\pm$ 0.9) | 4 bpp | 32 bpp |
| FedDQ | 91.0 ($\pm$ 0.2) | 3.6 bpp | 86.1 ($\pm$ 0.2) | 2.5 bpp | 74.0 ($\pm$ 0.4) | 2.6pp | 32 bpp |
| DAdaQuant | 90.9 ($\pm$ 0.3) | 3.1 bpp | 86.9 ($\pm$ 0.3) | 3.3 bpp | 74.4 ($\pm$ 0.9) | 3 bpp | 32 bpp |
| **FedBiF** | **92.7** ($\pm$ 0.1) | **1** bpp | **90.5** ($\pm$ 0.1) | **1** bpp | **79.8** ($\pm$ 0.7) | **1** bpp | **4** bpp |
| CIFAR-10 with ResNet-18 (8.3 GB) | | | | | | | |
| FedAvg | 75.2 ($\pm$ 0.2) | 32 bpp | 70.7 ($\pm$ 0.2) | 32 bpp | 63.2 ($\pm$ 0.3) | 32 bpp | 32 bpp |
| SignSGD | 71.7 ($\pm$ 0.3) | 1 bpp | 60.8 ($\pm$ 0.8) | 1 bpp | 52.9 ($\pm$ 1.9) | 1 bpp | 1 bpp |
| FedPAQ | 74.8 ($\pm$ 0.3) | 4 bpp | 70.3 ($\pm$ 0.4) | 4 bpp | 63.5 ($\pm$ 0.3) | 4 bpp | 32 bpp |
| FedDQ | 75.4 ($\pm$ 0.4) | 1.5 bpp | 70.3 ($\pm$ 0.4) | 2.3 bpp | 63.9 ($\pm$ 0.7) | 3.1 bpp | 32 bpp |
| DAdaQuant | 74.6 ($\pm$ 0.4) | 3.4 bpp | 68.9 ($\pm$ 0.3) | 3.3 bpp | 59.6 ($\pm$ 0.6) | 3.0 bpp | 32 bpp |
| **FedBiF** | **81.7** ($\pm$ 0.1) | **1** bpp | **77.7** ($\pm$ 0.1) | **1** bpp | **67.0** ($\pm$ 0.5) | **1** bpp | **4** bpp |
| CIFAR-100 with ResNet-34 (15.9 GB) | | | | | | | |
| FedAvg | 42.8 ($\pm$ 0.2) | 32 bpp | 40.6 ($\pm$ 0.3) | 32 bpp | 39.3 ($\pm$ 0.4) | 32 bpp | 32 bpp |
| SignSGD | 37.5 ($\pm$ 1.0) | 1 bpp | 34.9 ($\pm$ 1.4) | 1 bpp | 34.1 ($\pm$ 0.8) | 1 bpp | 1 bpp |
| FedPAQ | 42.5 ($\pm$ 0.3) | 4 bpp | 40.2 ($\pm$ 0.2) | 4 bpp | 39.7 ($\pm$ 0.3) | 4 bpp | 32 bpp |
| FedDQ | 43.4 ($\pm$ 0.4) | 1.4 bpp | 40.3 ($\pm$ 0.4) | 2.9 bpp | 39.4 ($\pm$ 0.4) | 2.5 bpp | 32 bpp |
| DAdaQuant | 42.5 ($\pm$ 0.6) | 3.3 bpp | 39.4 ($\pm$ 0.3) | 3.4 bpp | 36.9 ($\pm$ 0.2) | 3.3 bpp | 32 bpp |
| **FedBiF** | **50.6** ($\pm$ 0.3) | **1** bpp | **50.2** ($\pm$ 0.3) | **1** bpp | **46.3** ($\pm$ 0.2) | **1** bpp | **4** bpp |

In both IID and Non-IID settings, FedBiF consistently outperforms all baselines across all entries. SignSGD achieves 1 bpp communication, however as shown in Figure 2, suffers from slow convergence speed and poor test accuracy. Compared to FedPAQ, FedDQ and DAdaQuant do improve the communication efficiency by varying the bitwidth during training, however, the improvement is limited. Moreover, DAdaQuant converges slower due to the lower bitwidth used in the initial rounds. This issue is also reported in the original paper Hönig et al. (2022), further illustrating the limitations of post-training quantization at lower bandwidth. Unlike them, thanks to the proposed bits freezing, FedBiF achieves stronger compression meanwhile improving the accuracy by 2∼9%.

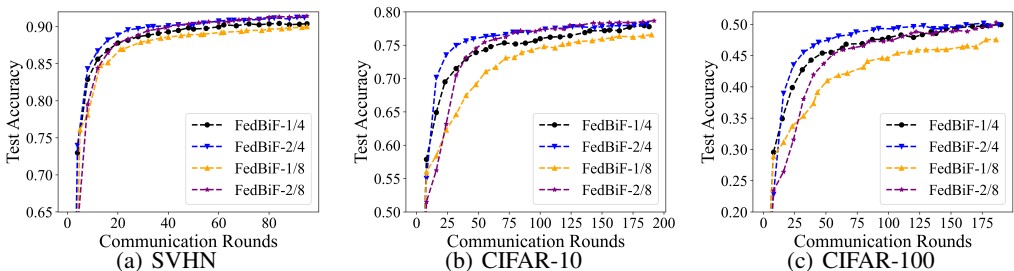

Figure 3: Convergence curves of FedBiF-$A/B$, where $A \in \{1, 2\}$ is the number of activated bits during each round and $B \in \{4, 8\}$ is the quantization bitwidth.

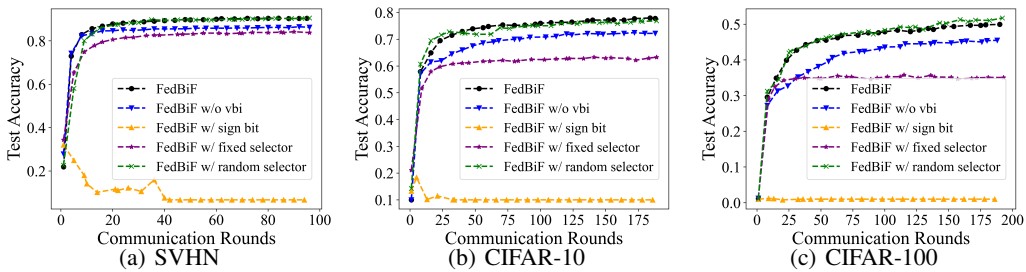

Figure 4: Convergence curves of various FedBiF variants, including with a sign bit, without virtual bits inheritance, with a fixed or random selector, respectively.

### 4.3 ABLATION STUDIES

In this part, we verify the scalability of FedBiF by varying the quantization bitwidth and the number of activated bits. Furthermore, we illustrate the effectiveness of FedBiF's design by evaluating various variants of FedBiF. All ablation experiments are conducted on Non-IID ($\mu = 0.5$) datasets.

**Scalability of FedBiF.** We use FedBiF-$A/B$ to denote the version of FedBiF in which the number of activated bits is $A$ and the quantization bitwidth is $B$. We vary $A$ in {1,2} and $B$ in {4,8}, respectively. As shown in Figure 3, FedBiF-1/8 exhibits a slower convergence speed and a lower accuracy than FedBiF-1/4. It is an intuitive observation as 8 rounds is required for FedBiF-1/8 to update all bits inside the model parameters, which is twice as that of FedBiF-1/4. Conversely, FedBiF-2/4 demonstrates the fastest convergence speed for the same reason. On the other hand, FedBiF-2/8 yields more precise representations thereby achieving better test accuracy than FedBiF-1/4.

**Effectiveness of FedBiF's design.** As shown in Figure 4, we evaluate various FedBiF variants, including without virtual bits inheritance (VBI), with a sign bit and with a fixed or random selector.

Firstly, without VBI, FedBiF obtains a lower accuracy, but still reaches that of FedAvg. We analyze that the reinitialized virtual bits require additional training to fit the local data and approach its data distribution of the previous round. The incorporation of VBI can effectively preserve local model features and provide valuable guidance for subsequent rounds of model training.

Secondly, with a sign bit, FedBiF fails to train a usable model. As shown in Figure 5, when training a sign bit in FedBiF, most model parameters decay to zero rapidly. The reason behind this phenomenon is that a change of the sign bit causes exactly the opposite value of the magnitude bits. For example, assuming a parameter $p = +m$ is sent to some clients with the same amount of data, where $+$ is the sign and $m$ is the sum of the magnitude bits. When training is performed on the sign bit and half of clients update the sign to $-$, the aggregated results of $p$ will be zero. Even if a small fraction of clients change the sign of $p$, the magnitudes of aggregated results will also decrease quickly.

Thirdly, with the fixed selector which always activates the third bit, FedBiF achieves a worse accuracy than that of FedAvg, which illustrates the necessity of alternately activating different bits. However, compared to SignSGD, FedBiF with the fixed selector achieves faster convergence and better accuracy, which proves the advantages of restricting model updates within low bitwidth during training. On the other hand, the random selector, activating a random bit in different rounds, achieves comparable

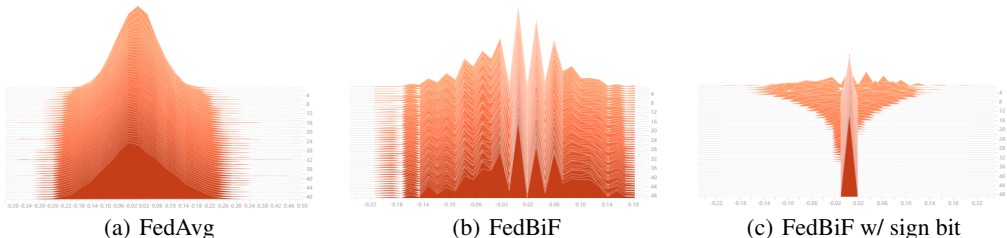

(a) FedAvg      (b) FedBiF      (c) FedBiF w/ sign bit

Figure 5: The weight distributions of the linear layer of ResNet-18 at the first 50 rounds.

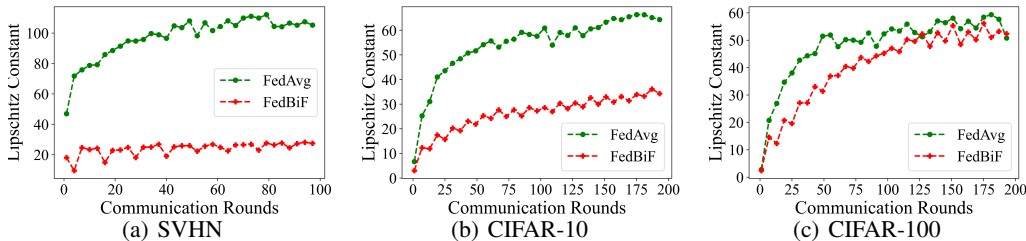

(a) SVHN      (b) CIFAR-10      (c) CIFAR-100

Figure 6: The Lipschitz constants of models trained by FedAvg and FedBiF.

accuracy as the default selector. Particularly, the random selector obtain slightly better accuracy on CIFAR-100 with a random bit sequence. This enlightens us that there may be better bit selection strategies, which we leave for future work.

### 4.4 BENEFICIAL PROPERTIES

In this subsection, we discuss two beneficial properties of FedBiF that we discovered experimentally.

**Better Generalization.** As presented in Section 4.2, FedBiF achieves better accuracy than FedAvg. To gain a deeper understanding of the accuracy improvement, we compare the Lipschitz constants of the models trained by FedAvg and FedBiF. Generally, Lipschitz constant is used to measure the Lipschitz continuity of nerual networks, where smaller Lipschitz constants represent better Lipschitz continuity and better generalizationVirmaux & Scaman (2018); Weng et al. (2018). We calculate the Lipschitz constant of the aggregated model in each round for both FedAvg and FedBiF. The definition of Lipschitz continuity and the calculation of Lipschitz constants can be found in Appendix C. As shown in Figure 6, FedBiF can restrict the Lipschitz constants of the trained global models to smaller values, which tends to produce better generalization capabilities.

**Model Sparsity.** Another benefit is that FedBiF can naturally generate sparse networks. In our experiments, FedBiF achieves an average sparsity of about 30% under different settings, see Appendix B.2 for detail. We analyze that a parameter will be pruned when its inside bits, except the first one, are optimized to zero. This enables FedBiF to easily prune redundant parameters during training, which we believe partly helps FedBiF achieve higher accuracy. Furthermore, the model sparsity may be utilized to further compress the downlink communication costs.

## 5 RELATED WORK

### 5.1 COMMUNICATION-EFFICIENT FEDERATED LEARNING

Existing methods primarily address the communication obstacle associated with FL in three ways, including infrequent aggregation algorithms, model compression and gradient compression.

The infrequent aggregation algorithms aim to reduce the number of rounds for FL to converge, thereby mitigating the communication overhead. FedAvg McMahan et al. (2017) is the most classic infrequent aggregation algorithm that performs multiple local optimization steps in each round. Also, FOLB Nguyen et al. (2021) performs intelligent sampling of clients to speed up convergence.

However, transferring large models between the server and the clients is still a significant burden, especially for the clients with limited communication bandwidth. To this end, researchers have turned to model compression techniques. Caldas et al. (2018); Bouacida et al. (2021) enable clients to train randomly selected subsets of a larger server model. FedPara Hyeon-Woo et al. (2022) shrink the model size by decomposing the weight matrices of layers into smaller matrices. Jiang et al. (2019); Li et al. (2021); Isik et al. (2023) adopt pruning techniques to get sparse networks for communication compression. It is worth noting that FedPM Isik et al. (2023) can achieve 1 bpp communication by only training and communicating a binary mask for each parameter. Similar to FedBiF but different, they freeze the entire weights at their initial random values and prune the weights by learning a binary mask. However, as reported in Vallapuram et al. (2022), freezing the weights at initial random values fails to reach the accuracy of FedAvg. Unlike them, the proposed framework FedBiF achieves 1 bpp communication by training only one bit inside a weight during each round. In this way, the weights can still be optimized by training each bit iteratively. Moreover, FedBiF can automatically prunes a weight when the bits inside the weight, except the first one, is optimized to zero.

Apart from the model compression, another way to reduce the communication costs per round is gradient (i.e., model update) compression, which is generally achieved by sparsification Rothchild et al. (2020); Qiu et al. (2022) or quantization Reisizadeh et al. (2020); Bernstein et al. (2019). Recent quantization methods Qu et al. (2022); Jhunjhunwala et al. (2021); Hönig et al. (2022) try to enhance compression ability by adjusting the quantization bitwidth adaptively during training. However, as discussed in Section 1, the main drawback of these methods lies in the post-training manner. The model updates are only quantized when local training finishes, so that the quantization errors of model parameters inevitably result in loss of model accuracy. FedBiF overcomes such drawback by training several activated bits inside each quantized parameter. The model updates are restricted within the representation of activated bits during local training, requiring no post-training quantization.

## 5.2 QUANTIZATION

The deep learning community has extensive research and applications on quantization methods, which can be roughly divided into two categories: post-training quantization (PTQ) Banner et al. (2019); Nagel et al. (2020) and quantization-aware training (QAT) Courbariaux et al. (2015); Choi et al. (2018); Esser et al. (2020). PTQ quantizes a pre-trained model while QAT quantizes the model during training. Specifically, QAT usually quantize the model parameters in the forward pass and update the full-precision parameters by STE Hinton (2012) during backpropagation. Extensive research Esser et al. (2020); Jacob et al. (2018) have proved that QAT usually enjoys better model accuracy than PTQ. Nevertheless, the quantization methods in federated learning Reisizadeh et al. (2020); Qu et al. (2022); Hönig et al. (2022) remain under the paradigm of PTQ. In this paper, we dispute this phenomenon and suggest that we should introduce the quantization of model updates into the local training process. To the best of our knowledge, FedBiF is the first effort to implement this functionality in FL. It is worth noting that the bit-wise training in Ivan (2022) has certain similarities with the proposed bits freezing. However, Ivan (2022) is designed to train a sparse network in centralized training. Also, we differ from them in the way of decomposing a weight and the training strategy.

## 6 CONCLUSION AND FUTURE WORK

In this paper, we investigate the problem of extensive communication overhead in federated learning via quantization techniques. We dispute the post-training manner of existing federated quantization methods and propose to restrict the model updates into fewer bits during local training. To this end, we propose FedBiF that enables clients to train and communicate partial bits inside parameters in each round. Extensive experiments validate not only the superiority of FedBiF in bidirectional communication compression but also the beneficial properties of model sparsity and better generalization.

The exciting results of FedBiF open new avenues for communication-efficient federated learning. We look forward to seeing more work introducing the quantization of model updates into the local training process. In addition, we notice that there are still some designs that can be further improved for FedBiF. For example, we believe that the bit selection strategy can be designed to achieve better convergence speed. Also, the step size can be designed to be learnable, which may further improve model accuracy and boost the FL model convergence. We leave these for future work.

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
