# OpenReview forum: "FedBiF: Communication-Efficient Federated Learning via Bits Freezing"
_ICLR.cc/2024/Conference — ICLR 2024 Conference Withdrawn Submission_

### Official Review · Reviewer_Gemw · 2023-10-26

**Soundness:** 2 fair
**Presentation:** 2 fair
**Contribution:** 2 fair
**Rating:** 3
**Confidence:** 4

**Summary:**

This paper considers the problem of communication efficiency in federated learning.

In such setups, iterative model and/or update transmissions occur between a central server and clients. These frequent exchanges of information result in a significant communication burden.

This paper proposes FEDBIF, a new technique for communication-efficient federated learning using bits freezing. Namely, in FEDBIF, the central server quantizes the model and sends it to the participating clients. Then, the clients perform local training on the quantized model, optimizing only chosen activated bits of the quantized model parameters (using virtual bits and STE). Finally, only the optimized bits are sent back to the server that aggregates the updates and updates the model parameters.

Comparison to several alternatives over three image classification tasks shows that FEDBIF performs well with less bandwidth (e.g., 4 bits for downlink and 1 bit for uplink) and even outperforms FedAVG.

**Strengths:**

1. Bi-directional communication efficiency in federated learning is timely and important.

2. Bit freezing is an interesting idea that allows for bi-directional compression.

3. Lowering the Lipschitz constant is an interesting and useful result.

**Weaknesses:**

1. The proposed method is completely empirical with no theoretical intuition or support or convergence guarantees.

2. The paper does not mention or compare to recent works that offer SOTA bi-directional compression capabilities. For example:

[1] Tyurin, Alexander, and Peter Richtárik. "2Direction: Theoretically Faster Distributed Training with Bidirectional Communication Compression." (NeurIPS 2023).

[2]  Kaja Gruntkowska, Alexander Tyurin and Peter Richtárik. "EF21-P and friends: Improved theoretical communication complexity for distributed optimization with bidirectional compression" (ICML 2023)

[3] Dorfman, R., Vargaftik, S., Ben-Itzhak, Y., & Levy, K. Y. "DoCoFL: Downlink Compression for Cross-Device Federated Learning." (ICML 2023)

3. The evaluation and considered baselines are not sufficient: (a) no comparison to SOTA as mentioned above. (b) the evaluation uses 10 clients where all clients participate in all rounds. Such a small number of clients with full participation is expected in cross-silo FL where updates can be sent in both directions and there is no need to quantize model parameters. (c) the reported baseline accuracies are very low suggesting that the hyperparameters are not tuned. E.g., for CIFAR10 and CIFAR100 the baseline accuracies are expected to be, e.g., 85+  and 65+ respectively (to the least and surely for the IID setup).  (d) For non-IID setups a good baseline would be one that takes the non-IID consideration into account. e.g., SCAFFOLD, FedProx.

**Questions:**

See weaknesses 1, 2 and 3.

---

### Official Review · Reviewer_eFqe · 2023-10-31

**Soundness:** 2 fair
**Presentation:** 3 good
**Contribution:** 2 fair
**Rating:** 3
**Confidence:** 4

**Summary:**

This paper presents FedBiF, a method that restricts the model updates within lower quantization bitwidth during local training. This is done by training partial individual bits inside each parameter, while freezing the others.

**Strengths:**

Originality:
The proposed work is moderately novel. The proposed bit freezing technique can become an interesting direction for future research.
When bits are viewed as quantizers of parameters, this can be thought of as a tree structure vector quantizer where some paths in tree are fixed and quantization update is limited to some paths in the tree.

Quality:
The quality of this work is fair.

Clarity:
The paper is well-organized, providing clear and concise explanations of the proposed method and its contributions.

Significance:
This work has the potential to inspire future research on freezing bits of parameters in training neural networks and expand on the relationship to other multilevel and tree structure quantizers.

**Weaknesses:**

1) Several claims made in the introduction section are not well justified.

a) The downlink model updates are compressed in existing works because such quantization is not needed as the aggregated model parameters are already in the quantized domain. b) Quantization aware training is well-known in the literature [1]. c) The proposed method does not completely eliminate quantization error either. It is debatable whether the proposed method outperforms post-training quantitation and quantization aware training.


2) Several key aspects of this paper are not well-developed. The paper claims FedBiF can outperform FedAvg because it achieves better generalization by limiting the growth of the model’s Lipschitz constant. The Lipschitz constant of other quantization methods are not provided in Figure 6. It is not clear whether it is related to quantization in general or FedBiF itself. Furthermore, I think the Lipschitz constant is small due to the model sparsity that may be introduced by FedBiF. However, to my best knowledge, pruning methods typically do not guarantee better generalization in deep networks. One evidence is that CIFAR10 with ResNet18 can easily achieve above 90\% accuracy [2] while the paper reports 75.2\%. Therefore, the optimization implementation may not be sound and the results in Table 1 needs further study. Finally, the choice of activated bits are rather random and worth further investigation. After all, no theoretical analysis is provided. My advice is to study bit freezing without Federated Learning (FL) and carefully study its behavior in optimization before merging it with FL.

3) I appreciate the author's acknowledgment of potential memory consumption issues in the appendix. Without their proposed trick of aggregating frozen bits, the memory consumption would become $m$ times that of the original local training. However, when bitwith increases, the memory consumption would also increase accordingly. Therefore, this limitation makes this method less attractive in practice as the local memory resources are very valuable.

4) There's a typo: "Sectoion 3.2" should be "Section 3.2".

[1] Training Deep Neural Networks with 8-bit Floating Point Numbers
[2] HeteroFL: Computation and Communication Efficient Federated Learning for Heterogeneous Clients

**Questions:**

1) What is recovery step in Figure 1? This is not discussed in the paper and it is not clear what it means.

---

### Official Review · Reviewer_Mh8E · 2023-10-31

**Soundness:** 3 good
**Presentation:** 3 good
**Contribution:** 3 good
**Rating:** 6
**Confidence:** 3

**Summary:**

The paper propose a Federated Learning (FL) communication bandwidth reduction method: FedBiF, via freeze parameters' bits during local training, which drastically reduces communication bandwidth for both uplink and downlink. In the meanwhile FedBiF achieves good performance on FL training (which is on par with FedDQ and FedPAQ in SVHN, while beat by large margin on CIFAR-10/100). The experiments shows that FedBiF could be pretty useful for FL training systems to save bandwidth, while doesn't sacrifice performance.

**Strengths:**

The method is composed the following components, which is examined via ablation study:

(1) subtract rather than signed integer, avoids all weights decay to 0, makes local model capable (training with sign just doesn't work).
(2) VBI turns out to be useful. Performance without VBI seems to be on par with FedDQ/FedPAQ on CIFAR-10/100 experiments, which shows that VBI utilizes the soft information which lead to better learning on quantized settings.
(3) random selector significantly work better than fixed selector, which makes sense .

The method targets to (a) compress both UL and DL communication, and (b) avoid post-training quantization behavior, which are both innovative, and results in good performance. The experiment is extensive and supportive on widely used FL datasets.

**Weaknesses:**

The comparison to existing method is extensive, but not self-contained, which makes reader hard to understand the difference between different methods.

**Questions:**

I am very curious about the result on Fig.4 and Fig.6. It seems that FedBiF without VBI has exact same performance compared against FedDQ and FedPAQ, while FedBiF with VBI outperforms on CIFAR-10/100 cases. It seems to me that VBI is a critical part of the design of algorithm. The explanation in the paper, seems to me is partial. A further comparison between methodology of FedDQ/FedPAQ and FedBiF could be very interesting.

---

### Official Review · Reviewer_t1mz · 2023-11-01

**Soundness:** 3 good
**Presentation:** 2 fair
**Contribution:** 2 fair
**Rating:** 3
**Confidence:** 4

**Summary:**

This work investigates communication-efficient federated learning, aiming to address the challenge of significant communication overhead during FL rounds. Specifically, the authors propose a new quantization-based compression method called bits freezing, which restricts the weight updates to a fewer bits. In each FL round, the model is updated only with a number of activated bits while freezing the others. The activated bit is selected in a circular manner to ensure the parameters of all bits are properly updated. Experimental results demonstrate the effectiveness of their method, showing not only reduction of communication costs but also improvement of generalization ability compared to FedAvg.

**Strengths:**

The proposed method exhibits promising results as the results not only show a reduction of communication overhead, but also improvement of the generalization performance compared to baselines.

The authors provide comprehensive ablation studies, which show the effectiveness of each component clearly.

**Weaknesses:**

All the experimental results are based on the setup with a small number of clients, i.e., 10 clients in this paper. I have concerns regarding the scalability and robustness of the proposed method to larger numbers of clients, such as $N=100$. It would be helpful to address this aspect.

The paper omits convergence guarantee, which raises doubts about the reliability of the method.

While the proposed method shows improved generalization performance compared to FedAvg, the paper lacks a detailed description or theoretical justification for why this happens. Why does FedBiF provide a lower Lipschitz constant?

**Questions:**

Could the authors provide the additional experimental results with a large number of clients? For example, consider 100 clients on CIFAR10 dataset. If the resources are limited, it is acceptable to evaluate it with a smaller model.

Why does FedBiF show better generalization performance than FedAvg even with quantization error? Why does FedBiF exhibit better Lipschitz continuity during FL?

Could the authors provide more detailed explanations on how VBI works? Which values are used to reinitialize the model before local update? More precisely, what does it mean by ‘original virtual bit’ in this step? Providing some concrete example would be helpful to understand.